# Research on Accurate Motion Trajectory Control Method of Four-Wheel Steering AGV Based on Stanley-PID Control

**DOI:** 10.3390/s23167219

**Published:** 2023-08-17

**Authors:** Weijie Fu, Yan Liu, Xinming Zhang

**Affiliations:** 1School of Mechatronic Engineering and Automation, Foshan University, Foshan 528001, China; weijie.fu@hotmail.com; 2School of Mechatronic Engineering, Changchun University of Science and Technology, Changchun 130022, China; 3Ministry of Education Key Laboratory for Cross-Scale Micro and Nano Manufacturing, Changchun 130022, China

**Keywords:** four-wheel-steering autonomous ground vehicle (4WS AGV), kinematics, motion trajectory control, Stanley-PID algorithm

## Abstract

With the continuous progress and application of robotics technology, the importance of mobile robots capable of adapting to specialized work environments is gaining prominence. Among them, achieving precise and stable control of AGVs (Automated Guided Vehicles) stands as a paramount task propelling the advancement of mobile robotics. Consequently, this study devises a control system that enables AGVs to attain stable and accurate motion through equipment connection and debugging, kinematic modeling of the four-wheel steering AGV, and a selection and comparative analysis of motion control algorithms. The effectiveness of the Stanley-PID control algorithm in guiding the motion of a four-wheel steering AGV is validated through MATLAB 2021a simulation software. The simulation results illustrate the outstanding stability and precise control capabilities of the Stanley-PID algorithm.

## 1. Introduction

With the emerging market of advanced Autonomous Ground Vehicles (AGVs) and mobile robots, the development of vehicle lateral and longitudinal control technology has become an important research topic. Research on AGVs primarily involves planning decisions, motion control, and environmental awareness, with motion control being the fundamental and crucial task for achieving precise AGV motion [1,2]. AGV motion control encompasses two key aspects: longitudinal control and lateral control [3,4,5,6]. Longitudinal control focuses on precise velocity control during AGV motion and employs methods such as PID control [1], model predictive control [2], and adaptive control [7]. PID control utilizes feedback information on velocity differences to adjust a vehicle’s acceleration, ensuring smooth, stable, and efficient motion. Lateral control involves correct path following and deviation correction, with approaches including Pure Pursuit control [8], LQR control [9], and Stanley control [10]. The Stanley control algorithm maintains the vehicle on the centerline by correcting forward errors, facilitating proper path tracking and deviation correction. For the motion control of a four-wheel steering AGV, the Stanley-PID algorithm integrates longitudinal and lateral control, optimizing control performance and achieving precise vehicle motion control while ensuring stability and reliability. By combining longitudinal control, lateral control, and optimization methods, the accuracy and reliability of AGV motion can be significantly improved.

Experts and scholars both domestically and internationally have conducted research on the motion trajectory control of Autonomous Ground Vehicles (AGVs), proposing adaptive control and fuzzy control methods [11,12,13]. However, adaptive control is prone to operational errors when sensors malfunction or behave abnormally, while fuzzy control rules cannot be adjusted online once determined, limiting their adaptability to changing situations [14]. Current research mainly focuses on nonlinear methods such as the theory of variable structure systems and fuzzy control for AGV motion control, particularly in the context of PID longitudinal control [15,16]. The complex composition and kinematic model of a four-wheel steering AGV can lead to inconsistencies in drive motor speeds during control, resulting in interference between motion units and issues such as vehicle swing or rollover [10]. The Stanley algorithm is suitable for guiding vehicles along pre-planned paths, but it may deviate from the original path when encountering deviations or unpredictable circumstances [15,17,18,19,20,21]. On the other hand, PID algorithms assist vehicles in making real-time adjustments, enabling them to maintain stability in various environments and conditions [15,16,18,22]. Based on the literature [1,2,7,9,11,12,13,14,15,16,18,21,22,23,24], it is evident that PID control algorithms are primarily applied in linear systems, while AGV motion control systems are nonlinear. PID algorithms mainly involve cumulative error calculations, making them susceptible to saturation. The Stanley algorithm focuses on lateral control and needs to be combined with other lateral control algorithms to achieve comprehensive motion control. Therefore, the Stanley-PID algorithm adjusts AGV vehicle speed and steering angle in real time by calculating error vectors and path deviations, ensuring that the AGV moves along a reasonable trajectory and accurately tracks the desired route, guaranteeing stability and precision.

This study aims to achieve precise motion control of a four-wheel steering AGV by combining longitudinal and lateral control methods, providing an innovative solution for AGV motion control and overcoming limitations of traditional control methods. By integrating longitudinal control, lateral control, and optimization techniques, the motion performance of AGVs in complex environments is improved. This paper conducts simulation analysis on the kinematic model of a four-wheel steering AGV and proposes a simulation experiment using the Stanley-PID algorithm for AGV motion control. A comparison is made with a single PID algorithm and the Stanley algorithm, demonstrating the effectiveness and feasibility of the Stanley-PID algorithm. By considering both longitudinal and lateral control and employing optimization methods, this research offers an innovative solution for AGV motion control, overcoming the limitations of traditional control methods, and enhancing the motion performance of AGVs in complex environments. The experimental results indicate the significant advantages of the Stanley-PID algorithm in achieving precise motion control, providing a viable solution for the practical application of AGVs.

## 2. Structure of the Motion Control System

The 4WS AGV platform consists of a motion chassis, four steering wheel mechanisms, and four lifting mechanisms. The motion chassis is used to install the motion controller, batteries, servo drivers, and sensors, among other components. The structure of the motion control system for the AGV is illustrated in Figure 1. To achieve real-time coordinated control of the 12 servo motors, a CAN (Controller Area Network) bus is employed for communication between the controller and the drivers.

The four driving servo drivers and four lifting servo drivers utilize feedback signals from Hall sensors and incremental encoders mounted on the servo motors to achieve precise speed closed-loop control. In addition to monitoring the speed feedback signals, the four steering servo drivers also measure the rotation angle of the steering motors, enabling accurate position closed-loop control.

## 3. Kinematic Modeling

The AGV belongs to the category of nonholonomic motion systems. Unlike holonomic kinematics models, the nonholonomic kinematics model considers the effects of both dynamic and kinematic constraints on the motion state. The following assumptions are made [25,26]:The AGV is treated as a rigid body.The steering motors of each steering wheel are perpendicular to the ground, while the driving motors are parallel to the ground.The steering wheels only experience pure rolling without any relative sliding.

From a control perspective, there are three controlled variables for the steering wheels: *v* (appropriate velocity), *θ* (current steering angle of the steering wheel as a whole), and ω (rotational angular velocity). The absolute coordinate system (*X*_0_, *Y*_0_, *Z*_0_) and the relative coordinate system (*X_OI_*, *Y_OI_*, *Z_OI_*) are defined, where the geometric center of the 4WS AGV vehicle coincides with the origin of the relative coordinate system. The *Y*_1_-axis represents the lateral direction, and the *X*_1_-axis represents the longitudinal direction. The angle *θ* represents the orientation between the absolute and relative coordinate systems. The main motion parameters of the AGV are summarized in Table 1.

In order to simplify the complexity of the inverse kinematics, the AGV vehicle is selected to perform a curved motion around a point in the relative coordinate system. The motion is illustrated in Figure 2. The angle between the velocity direction at the centroid of the AGV and *Y_I_* is denoted as *θ_I_* = arctan(*b*/*a*), while the angle between the velocity direction at the centroid and Y0I is denoted as *θ_0_* = *θ_I_* + *θ*. The steering angles of the four wheels are represented as *α* = *β* = *η* = *γ*, and the velocity relationship is denoted as *V*_1_ = *V*_2_ = *V*_3_ = *V*_4_. Based on the geometric kinematics model [16], the pose of the AGV centroid in the absolute coordinate system can be represented as $\dot{X}={\left[\begin{array}{ccc} \dot{x} & \dot{y} & \dot{\varphi } \end{array}\right]}^{T}$.

Based on Figure 2b:(1)$$\left\{\begin{array}{c} a=\frac{\left[\sin\alpha \cos\beta x_{1}-\cos\alpha \sin\beta x_{2}-\cos\alpha \cos\beta \left(y_{1}-y_{2}\right)\right]}{\sin\left(\alpha -\beta \right)}\\ b=\frac{\left[\sin\alpha \cos\beta \left(x_{1}-x_{2}\right)-\cos\alpha \sin\beta y_{2}-\cos\alpha \cos\beta y_{2}\right]}{\sin\left(\alpha -\beta \right)} \end{array}\right.$$

The radius from each steering wheel to the center of rotation *O_I_* can be expressed as:(2)$$\left\{\begin{array}{c} R_{i}=\sqrt{{\left(y_{i}-b\right)}^{2}-{\left(a-x_{i}\right)}^{2}}\\ R=\sqrt{a^{2}-b^{2}} \end{array}\;\;\;\;i=1,2,3,4\right.$$

The angular velocity at the rotation center *O_I_* is *ω* = *V_i_*/*R_i_*, and the velocity direction of the AGV in the absolute coordinate system is *θ*_0_ = *θ_I_* + *θ*. Here, $|x_{1}|=|x_{2}|=|x_{3}|=|x_{4}|=D/2$, $|y_{1}|=|y_{2}|=|y_{3}|=|y_{4}|=L/2$, and the rotation angle of each steering wheel can be calculated as [16]:(3)$$\left\{\begin{array}{c} \alpha =\tan^{-1}\left[\left(y_{1}-b\right)/\left(x_{1}-a\right)\right]\\ \beta =\tan^{-1}\left[\left(y_{2}-b\right)/\left(x_{2}-a\right)\right]\\ \eta =-\tan^{-1}\left[\left(-y_{3}+b\right)/\left(x_{3}-a\right)\right]\\ \gamma =-\tan^{-1}\left[\left(-y_{4}+b\right)/\left(x_{4}-a\right)\right] \end{array}\right.$$

In Equations (1)–(3), the parameters (driving speed and rotation angle) of each steering wheel can be solved, and the AGV can walk and turn according to the target trajectory [15,16,22,24].

By performing a MATLAB 2021a software simulation with a specified size of 7000 mm × 3590 mm × 531 mm for the 4WS AGV and a moving speed of *V_I_* = 0.05 m/s, the inverse kinematics are used to calculate the velocities and angles of the four steering wheels. The results are presented in Table 2.

The forward kinematics model of the AGV vehicle determines the velocities and angles of each steering wheel. The inverse kinematics simulation results are used as inputs to validate the forward kinematics simulation parameters, as shown in Figure 3.

The AGV vehicle follows a circular path with (0, 0) as the center, as shown in Figure 3c. Within 60 s, the AGV vehicle exhibits sinusoidal and cosine variations along the *X* and *Y* axes, respectively, ranging between (−1, 1) and (−2, 0). After performing simulation calculations, a rotational radius of 1 m and a linear velocity of 0.1 m/s are obtained. The simulation, combining both forward and inverse kinematics models, mutually verifies their correctness.

## 4. Motion Trajectory Control Analysis

The AGV vehicle is a nonholonomic and nonlinear system, which requires an appropriate control method. The PID algorithm has the advantages of simple structure and easy implementation, but it can only work under linear conditions, leading to reduced control accuracy. The Stanley-PID control combines the advantages of the Stanley algorithm and the PID controller. It uses the Stanley algorithm to calculate the lateral error and feeds it into the PID controller to generate longitudinal commands. This approach can better adapt to dynamic characteristics and disturbances, resulting in more precise path-tracking control. By combining it with the trial-and-error method for tuning PID parameters, further improvement in control accuracy can be achieved. Table 3 illustrates the advantages and disadvantages of various controllers [27].

### 4.1. Stanley Control

The Stanley algorithm is a control method that takes into account the lateral error (*e*) between the AGV and the desired motion trajectory. It aims to minimize the lateral error and also considers the heading deviation (*θ_e_*) between the AGV’s orientation and the desired path. By utilizing the geometric relationship between the vehicle and the road during motion, the Stanley algorithm obtains the necessary variables for controlling the AGV’s movement, as shown in Figure 4 [14,15,21,35,36].

Based on the relative geometric relationship between the AGV’s motion pose and the given path, the control variables for controlling the AGV’s steering angle can be directly obtained. With the lateral error (*e*) and heading error (*θ_e_*), the steering angle for the steering wheels can be determined.
(4)$$\delta \left(t\right)=\theta_{e}\left(t\right)+\delta_{e}\left(t\right)$$

The rate of change of lateral error is denoted as:(5)$$\dot{e}\left(t\right)=-V\left(t\right)\sin\delta_{e}\left(t\right)$$

When *θ_e_* = *δ_e_*_,_ the AGV turns based on the heading error angle *θ_e_* to track the desired trajectory. Without considering the heading error angle *θ_e_*, assuming that the expected trajectory of the AGV intersects with the nearest tangent to the given path at a distance of *d*(*t*) from the steering wheel, the non-linearity proportional function can be derived based on the AGV’s motion geometry [20].
(6)$$\delta_{e}\left(t\right)=\arctan\frac{e\left(t\right)}{d\left(t\right)}=\arctan\frac{ke\left(t\right)}{d\left(t\right)}$$
where *d*(*t*) = *V*(*t*)/*k* and *k* are control gain parameters, and *d*(*t*) is the forward distance of the steering wheel, which is proportional to the speed of the steering wheel. Taking into account the two scenarios mentioned above, the basic steering angle control equation can be represented as follows:(7)$$\delta \left(t\right)=\theta_{e}\left(t\right)+\arctan\left[ke\left(t\right)/V\left(t\right)\right]$$

According to Figure 4, sin*δ_e_*(*t*) can be expressed as:(8)$$\sin\delta_{e}\left(t\right)=\frac{e\left(t\right)}{\sqrt{{\left(d\left(t\right)\right)}^{2}+{\left(e\left(t\right)\right)}^{2}}}=\frac{ke\left(t\right)}{\sqrt{{\left(V\left(t\right)\right)}^{2}+{\left(ke\left(t\right)\right)}^{2}}}$$

As shown in Equations (4)–(8), it can be derived that the rate of change of lateral error, *e*(*t*), is [15]:(9)$$\dot{e}\left(t\right)=-\frac{ke\left(t\right)}{\sqrt{1+{\left(\frac{ke\left(t\right)}{V\left(t\right)}\right)}^{e}}}$$

In Equations (7) and (9), it can be observed that when the lateral deviation (*e*) is small or nonexistent, *δ*(*t*) = *θ_e_*(*t*); and when the heading angle (*θ_e_*) deviation is small or nonexistent, $\dot{e}\left(t\right)\approx -ke(t)$; the heading angle control equation can be expressed as:(10)$$\delta \left(t\right)=\arctan\left[ke\left(t\right)/V\left(t\right)\right]$$

Equation (10) is solved by applying the Laplace transform to the differential equation, resulting in:(11)$$\dot{e}=e\left(0\right)\times e^{-kt}$$

Therefore, the lateral error converges exponentially to *e*(*t*) = 0, where the parameter *k* determines the convergence rate. For any lateral error, the differential equation monotonically converges to 0.

The Stanley algorithm is a control strategy based on path tracking and local perception to achieve high-precision vehicle motion control. However, the Stanley algorithm is primarily suitable for vehicles driving on smooth road surfaces and has limitations when it comes to controlling AGVs that consider nonholonomic constraints and dynamic characteristics. The Stanley algorithm requires accurate initial position and orientation information as well as a smooth path. In complex trajectories and environments, the Stanley algorithm may fail. In areas such as tunnels and basements, the sensors may not provide reliable data.

### 4.2. PID Control

PID control is essentially a control strategy where the proportional coefficient reflects the current state of the input signal, the integral signal represents the cumulative changes of the signal, and the derivative signal indicates the trend of the input signal’s changes, as shown in Figure 5 [11,20].

Where *r*(*t*) represents the input of the control system, *u*(*t*) represents the output of the control system, *y*(*t*) represents the current value of the controlled object, and *e*(*t*) represents the error between the current system input *r*(*t*) and the desired output *y*(*t*). By calculating the deviation between the desired *r*(*t*) and the current output *y*(*t*), and integrating the proportional (*P*), integral (*I*), and derivative (*D*) components, the closed-loop control output is obtained to adjust the output of the controlled object. The PID control adopts the typical PID incremental control, and its expression is as follows:(12)$$u\left(k\right)=u\left(k-1\right)+K_{P}\left[e\left(k\right)-e\left(k-1\right)\right]+K_{I}e\left(k\right)+K_{D}\left[e\left(k\right)-2e\left(k-1\right)\right]+e\left(k-2\right)$$
(13)$$e\left(t\right)=r\left(t\right)-y\left(t\right)$$

In the equation, *K_P_* represents the proportional coefficient, *K_I_* (*K_I_* = *K_P_/T_I_*) represents the integral coefficient, and *K_D_* (*K_D_* = *K_P_*∙*T_D_*) represents the derivative coefficient.

In the motion trajectory control analysis of a 4WS AGV vehicle, the main reason for not directly using the Stanley algorithm is that its handling of the contour line is not suitable for AGV vehicles, which belong to a special type of nonholonomic kinematics systems. Therefore, in practical applications, it is common to combine other control algorithms and techniques to achieve more stable and reliable AGV motion trajectory control.

### 4.3. Parameter Tuning

Ref. [19] shows how Snider implemented the Stanley controller in Equation (7) and compared its performance in different motion environments.

The trial-and-error method [17] is a process of simulating or physically operating a closed-loop system, observing the system response curve, and repeatedly adjusting the parameters based on the approximate impact of each tuning parameter on the system response. This process is conducted to achieve a satisfactory response and determine the three tuning parameters in the PID control. MATLAB Simulink functionality is utilized to tune the PID parameters, as shown in Figure 6.

During the motion trajectory control of AGV, the main role of the PID algorithm is to calculate the control signal that enables the AGV to follow the expected trajectory. The trial-and-error method helps find suitable PID algorithm parameters to achieve more accurate and stable control. In practical applications, different control systems, control objects, and control requirements may lead to different choices of PID parameters.

### 4.4. Stanley-PID Control

The motion trajectory control of a four-wheel steering AGV (4WS AGV) belongs to a complex system. The combination of PID and Stanley control allows for the adjustment of both the longitudinal and lateral deviations. The control principle is illustrated in Figure 7.

According to Equation (13), the path deviation under Stanley-PID control is denoted as *e*(*t*) = *r*(*t*) − *y*(*t*). According to Equations (7) and (12), the steering wheel rotation angle *U*(*t*) under Stanley-PID control can be calculated as:(14)$$U\left(t\right)=\delta_{stanley}+K_{P}\left[e\left(k\right)-e\left(k-1\right)\right]+K_{I}e\left(k\right)+K_{D}\left[e\left(k\right)-2e\left(k-1\right)+e\left(k-2\right)\right]$$

In conclusion, selecting the appropriate control algorithm is crucial in analyzing AGV motion trajectory control. For applications requiring high precision control, the Stanley-PID control algorithm is worth considering as a preferred method due to its adaptability to complex road conditions.

### 4.5. Simulation Results Analysis

In the simulation setup, the AGV’s initial coordinates are (0, 0), with a velocity of 0 and a heading angle of 0. The sampling time is *T* = 60 s, the scan period is *ts* = 0.005 s, and the interval time is *dt* = 0.1 s. The desired value *r* is set to 1, and the simulation is performed using Stanley, PID, and Stanley-PID control methods.

For Stanley control, the motion velocities are set to 3 m/s, 2 m/s, 1 m/s, 0.5 m/s, and 0.3 m/s, with a control gain of 1 for the simulation.

The simulation results are shown in Figure 8. As the velocity increases under Stanley control, the AGV exhibits deviation, and the deviation becomes larger with higher velocities. On the other hand, as the velocity decreases, the vehicle experiences shaking.

In PID control, based on the existing forward and inverse kinematics models, and using the trial-and-error method, the following parameters are obtained: *K_P_* = 0.22, *K_I_* = 0.13, *K_D_* = 0.

When the initial position deviation of the AGV is the same, the deviation under PID control decreases rapidly. However, under PID control, there are 3 to 4 fluctuations in the motion trajectory, and the AGV vehicle converges to the desired trajectory after approximately 6 s, as shown in Figure 9.

Figure 10 represents the simulation analysis of Stanley-PID motion trajectory control. The initial state of the AGV vehicle is set as (0, 0, 0), and the desired running speed is 8 m/s. The lateral deviation is calculated using MATLAB 2021a software.

Figure 10 presents the simulated results of the fitted sinusoidal trajectory and rotation angle. The positive values (1.361) represent left steering wheel output, while the negative values (−0.648, −1.253) represent right steering wheel output. By combining the simulation results from Figure 8 and Figure 9, along with the error curve in Figure 11, it can be observed that under PID control, there is initially a fluctuation in position deviation, which is approximately 0.04. In comparison, Stanley-PID control exhibits smaller motion trajectory fluctuations and a more stable response.

Based on the comparison of the simulation results mentioned above, in the motion trajectory control of the AGV, Stanley-PID control exhibits faster response, smaller fluctuations, and smoother variations compared to a single PID control. This demonstrates stronger system stability.

## 5. Discussion

This study primarily focuses on equipment connection and debugging, kinematic modeling, as well as a comparative analysis of motion control algorithms for the four-wheel steering AGV (Automated Guided Vehicle). Firstly, we meticulously conduct equipment connection and debugging for the four-wheel steering AGV to ensure seamless communication and data transmission between the controller, driver, and steering devices. This crucial step lays a solid foundation for subsequent experiments and simulations. Secondly, a comprehensive kinematic model of the four-wheel steering AGV is established, thoroughly analyzing its motion characteristics and key parameters for precise modeling. This serves as the cornerstone for the design of control algorithms and motion simulations. Subsequently, our study delves into the selection of suitable control algorithms. We thoroughly compare the performance of the Stanley-PID algorithm, traditional PID algorithm, and Stanley algorithm in path-tracking tasks. The innovative Stanley-PID algorithm, which ingeniously combines the strengths of Stanley and PID controllers, calculates lateral errors using the Stanley algorithm and inputs them into the PID controller to generate longitudinal commands, resulting in unparalleled accuracy for path tracking control. Lastly, we thoughtfully summarize the aforementioned findings, extensively discuss factors impacting the motion control precision of the four-wheel steering AGV, and propose effective solutions to address these challenges. Our research endeavors to elevate the motion precision of the four-wheel steering AGV, offering robust and precise motion control solutions for practical applications. However, one limitation of this study lies in the lack of experimental validation. As a specialized mobile robot platform with distinct advantages, conducting experimental tests on the four-wheel steering AGV is essential for evaluating its overall performance. Through these experiments, comprehensive assessments can be made regarding its navigation accuracy, stability, energy consumption, and other performance indicators, while also facilitating comparisons with other types of AGVs. Without such experiments, the potential strengths and limitations of the research work may not be fully understood and demonstrated. Conducting experiments on the four-wheel steering AGV will allow for a better exploration and confirmation of these advantages, providing further references and directions for future research improvements.

## 6. Conclusions

In this study, we established the motion model of a four-wheel-steering AGV (4WS AGV) and introduced the principles of Stanley-PID control. Simulation analysis was conducted using MATLAB software. The results of the simulation experiments demonstrated that the Stanley-PID algorithm outperforms traditional PID algorithms, particularly in terms of tracking accuracy and steady-state error. In the analysis of motion trajectory control for the 4WS AGV, the Stanley-PID algorithm exhibited a response time of around 0.2, with the path error initially fluctuating between 0 and −1.5 and eventually stabilizing at around 0.4. On the other hand, the single PID algorithm had a response time of around 0.3, with the path error fluctuating between 0 and −1.6, thus confirming its effectiveness and feasibility. In the motion trajectory control of the 4WS AGV, the Stanley-PID control demonstrated advantages such as faster response, smaller fluctuations, smoother variations, and stronger system stability compared to single PID control.

Regarding the future development directions for the motion trajectory control of four-wheel steering AGVs, there are several aspects that deserve our attention:(1)Conducting four-wheel steering AGV experiments: To comprehensively assess the navigation and maneuvering performance of four-wheel steering AGVs, conducting a series of experiments to validate their capabilities is recommended. These experiments can be designed to test the behavior and performance of four-wheel steering AGVs in various scenarios and tasks.(2)Exploring path planning algorithms for four-wheel steering AGVs: Leveraging the maneuverability of four-wheel steering AGVs, further exploration and research on new path planning algorithms and optimization methods are worthwhile.(3)Exploring diverse application areas for four-wheel steering AGVs: Apart from traditional material handling and warehousing, consider applying four-wheel steering AGVs in indoor navigation, smart warehousing, automated logistics, and service robotics. Expanding the range of application areas will provide insights into the potential and prospects of four-wheel steering AGVs.(4)Technology integration: Integrating four-wheel steering AGVs with other technologies, such as sensor fusion, can achieve more precise environmental perception and autonomous navigation. Additionally, combining artificial intelligence, data analytics, and decision algorithms can enhance the intelligence level of four-wheel steering AGVs in complex environments.(5)Anomaly detection and robust control: With continuous technological advancements, the research of systems that would enable four-wheel steering AGVs to automatically detect and resolve faults is necessary. Designing robust motion trajectory control algorithms to maintain stable and reliable trajectory tracking capability under uncertainty and disturbances is also crucial.(6)Energy optimization: The Exploration of energy optimization methods for the motion trajectory control of four-wheel steering AGVs, aiming to prolong battery life, improve energy utilization efficiency, reduce dependence on charging devices, and achieve sustainable development with genuine energy-saving and emission reduction benefits is recommended.

## Figures and Tables

**Figure 1 sensors-23-07219-f001:**
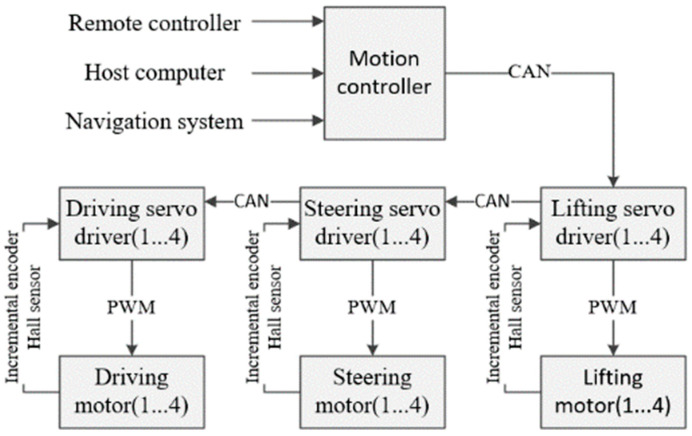
Structure of AGV Motion Control System.

**Figure 2 sensors-23-07219-f002:**
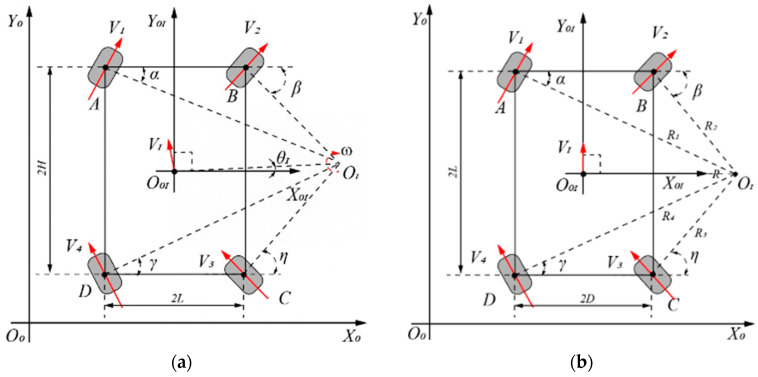
AGV vehicle motion is illustrated. (**a**) Forward kinematics (the rotation center does not lie on the *x*-axis of the relative coordinate system). (**b**) Inverse kinematics (the rotation center lies on the *x*-axis of the relative coordinate system).

**Figure 3 sensors-23-07219-f003:**
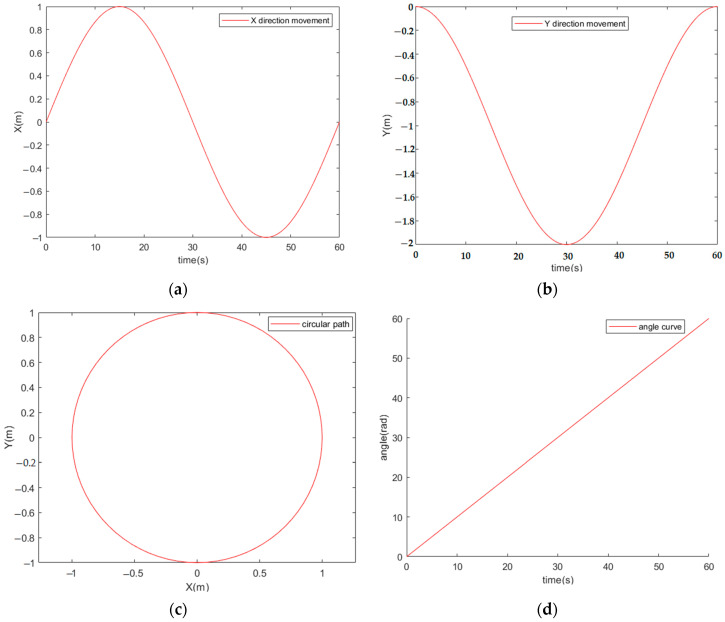
Simulation of AGV vehicle’s displacement in the *X* and *Y* axes, motion trajectory, and angular velocity. (**a**) Displacement in the *X*-axis; (**b**) Displacement in the *Y*-axis; (**c**) Circular motion trajectory; (**d**) Angular velocity.

**Figure 4 sensors-23-07219-f004:**
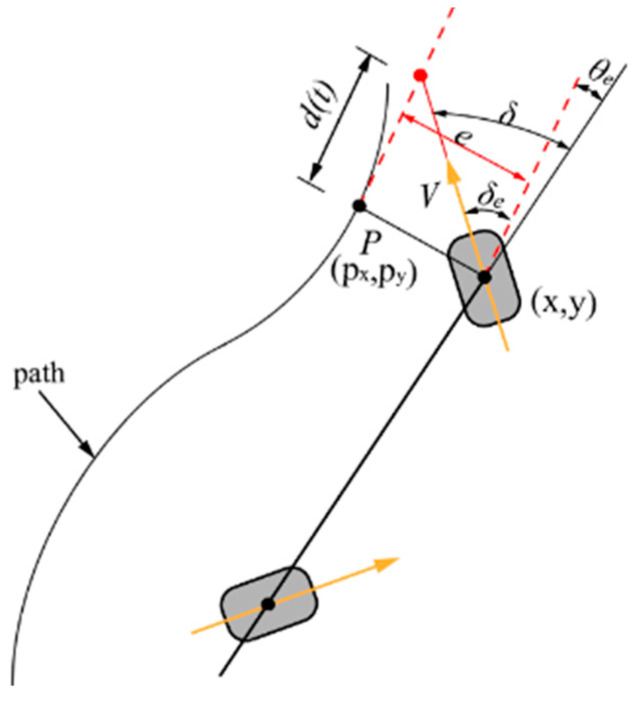
Illustration of Stanley Algorithm for AGV Vehicles.

**Figure 5 sensors-23-07219-f005:**
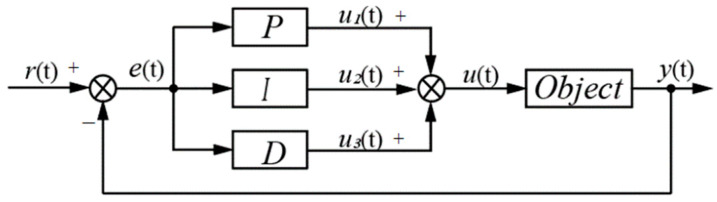
Schematic Diagram of PID Controller.

**Figure 6 sensors-23-07219-f006:**
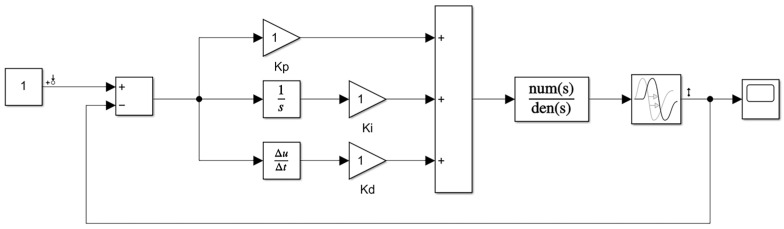
Trial-and-Error Method for PID Parameter Tuning.

**Figure 7 sensors-23-07219-f007:**
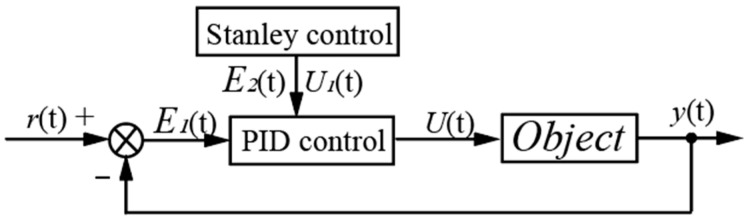
Stanley-PID Control Principle.

**Figure 8 sensors-23-07219-f008:**
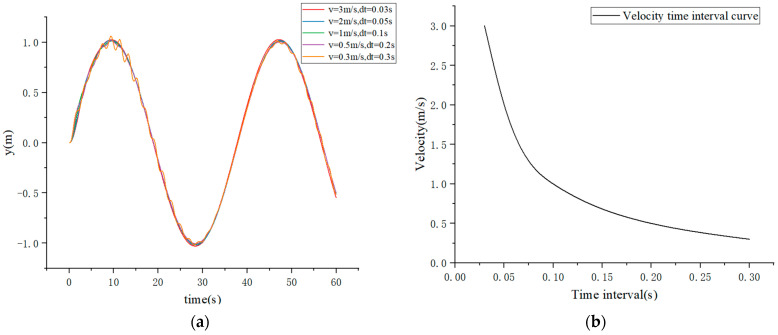
(**a**) Simulation Results of Stanley Control; (**b**) Velocity Time Interval Curve. The time interval refers to the time gap between updating the AGV state and plotting the path/vehicle status in each cycle.

**Figure 9 sensors-23-07219-f009:**
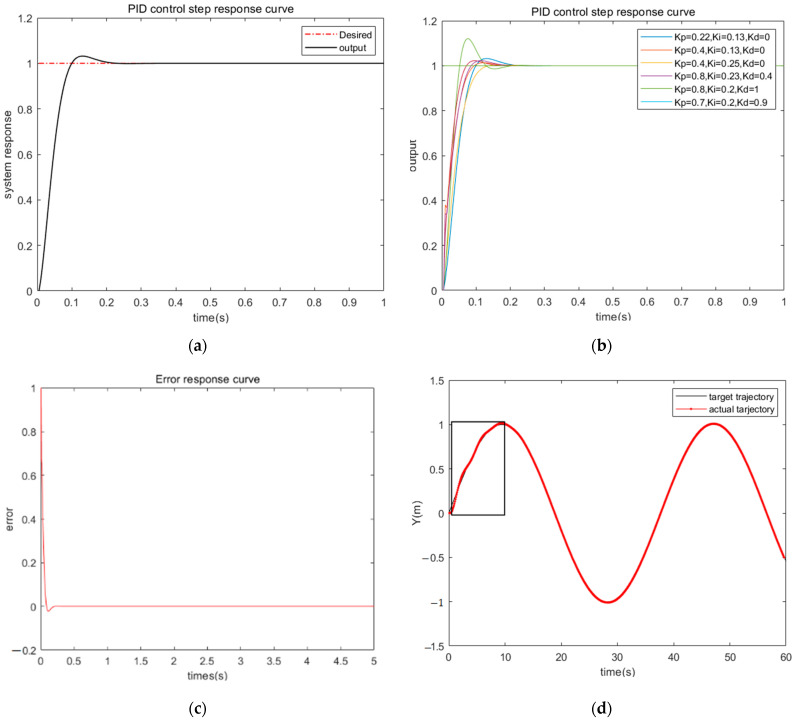
Simulation Results of PID Control (**a**) Step Response Curve; (**b**) Comparative Step Response Curve; (**c**) Error Response Curve; (**d**) Sinusoidal Tracking.

**Figure 10 sensors-23-07219-f010:**
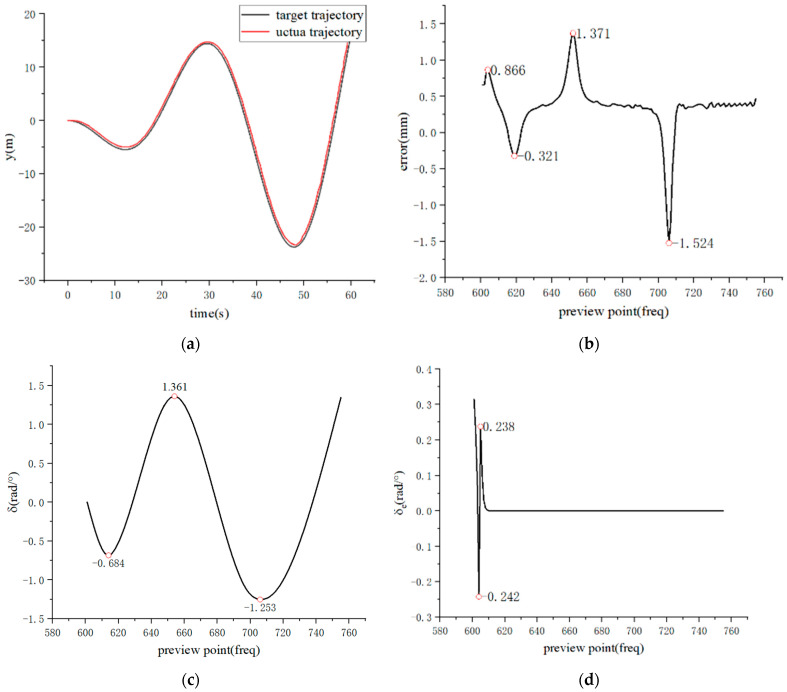
Simulation Analysis of Stanley-PID Motion Trajectory Control. (**a**) Sinusoidal Tracking Curve; (**b**) Path Error Curve; (**c**) Rotation Angle Curve; (**d**) Error Response Curve.

**Figure 11 sensors-23-07219-f011:**
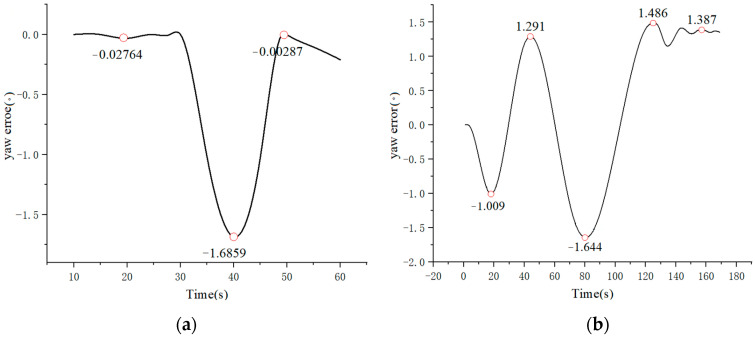
Error Curve. (**a**) PID Error; (**b**) Stanley Error.

**Table 1 sensors-23-07219-t001:** The main motion parameters of the AGV.

Symbols	Description	Symbols	Description
*A* (*x*_1_,*y*_1_)	Left front steering wheel	*V* _1_	Velocity of the left front wheel
*B* (*x*_2_,*y*_2_)	Right front steering wheel	*V* _2_	Velocity of the right front wheel
*C* (*x*_3_,*y*_3_)	Left rear steering wheel	*V* _3_	Velocity of the left rear wheel
*D* (*x*_4_,*y*_5_)	Right rear steering wheel	*V* _4_	Velocity of the right rear wheel
*θ*	Current steering angle of the steering system as a whole	*V_I_*	Speed of the centroid
*θ_I_*	Angle between the velocity direction of the centroid and *Y*_1_	*ω*	Angular velocity of the vehicle’s center point
*α*	Steering angle of the left front wheel	*2H*	Lateral axle distance
*β*	The steering angle of the right front wheel	*2L*	Longitudinal axle distance
*γ*	Steering angle of the left rear wheel	*O_I_*	Rotation center of the vehicle
*η*	Steering angle of the right rear wheel	*I*	Distance from the steering wheel to the center point of the vehicle
*V_ix_*	Longitudinal velocity of the steering wheels	*R*	Radius from the steering wheel to the rotation center
*V_iy_*	Lateral velocity of the steering wheels	*R_i_*	Radii from the four steering wheels to the rotation center
*a*	Lateral coordinate of the turning center	*b*	Lateral coordinate of the rotation center

**Table 2 sensors-23-07219-t002:** AGV vehicle inverse kinematics simulation results.

	Left Front Steering Wheel 1	Right Front Steering Wheel 2	Right Rear Steering Wheel 3	Left Rear Steering Wheel 4
Angle (rad)	0.33795	−0.6227	−0.3795	0.6227
velocity (m/s)	0.2422	0.1539	0.2422	0.1539

**Table 3 sensors-23-07219-t003:** Comparison of Controller Advantages and Disadvantages.

Controller Type	Advantages	Disadvantages
Linear Quadratic Regulator (LQR) [28]	Strong Stability and Robustness.	Overly dependent on system models and not suitable for nonlinear systems; complex computation and storage for large-scale systems.
Model Predictive Control (MPC) [29]	More suitable for path tracking.	Complex design and higher cost.
Pure Pursuit Control (PPC) [30]	No need for model prediction and algorithm optimization; fast response; applicable to various systems.	Existence of static errors without self-correction capability.
Sliding Mode Control (SMC) [31,32]	Good robustness; low system model requirement.	Measurement errors have significant impact on its performance; Exhibits oscillations.
PID control [11,14,20]	Simple computation; strong adaptability; fast response speed; strong path tracking ability; good robustness; real-time and easy-to-implement performance.	Unable to adapt to variations in system parameters.
Stanley control [10,33,34]	Simple computation; strong adaptability; fast response speed; strong path tracking ability; good robustness; real-time and easy-to-implement performance.	Specific scenarios and requirements need to be considered.

## Data Availability

Not applicable.

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
