# Peer review of "Research on Accurate Motion Trajectory Control Method of Four-Wheel Steering AGV Based on Stanley-PID Control"

_sensors, 2023, doi:10.3390/s23167219_

Round 1

Reviewer 1 Report

This manuscript investigates the precise motion trajectory control method of a 4-wheel steering autonomous ground vehicle (4WS AGV) based on Stanley-PID control. However, there are still the following problems.

1. What is the purpose of combining Stanley algorithm and PID control algorithm in this manuscript? What is the main work carried out by the author here? What are the difficult issues encountered?

2. What are the advantages of the research content in the manuscript compared to other similar articles?

3. In the manuscript, the abstract needs to refine the language in order to better highlight the main work of the manuscript.

4. If the assumptions in the manuscripts are proposed by the author himself, please explain their rationality, otherwise please cite the corresponding literature.

5. There are certain formatting problems in the manuscript, such as the size of formulas, etc.

6. The format of the references is not uniform, and in addition, it is necessary to cite some of the latest high-level literature.

Minor editing of English language required

Author Response

Dear Reviewer and Editor,

Thank you for giving us the opportunity to submit a revised draft of the manuscript “Research on Accurate Motion Trajectory Control Method of 4-Wheel Steering AGV Based on Stanley-PID Control (sensors-2542127)” for publication in the Journal of Sensors. I appreciate the time and effort that you and the reviewers dedicated to providing feedback on my manuscript. I are grateful for the thoughtful comments on the manuscript. I have incorporated changes to reflect most of the suggestions made by the reviewers. I have highlighted the changes within the revised manuscript.

In response to these comments, I have made the following changes:

Response to Reviewer 1

Q1: What is the purpose of combining Stanley algorithm and PID control algorithm in this manuscript? What is the main work carried out by the author here? What are the difficult issues encountered?

Answer 1: Thank you very much for your valuable review comments and inquiries. We greatly appreciate your attention to the research objectives, main contributions, and challenges encountered.

  1. a) Objectives: The main objective of this study is to achieve precise motion trajectory control for the 4-wheel steering AGV. Traditional PID control algorithms may exhibit certain trajectory deviation issues in AGV control, while the introduction of the Stanley algorithm allows for more accurate tracking of the designated motion trajectory, thereby enhancing control precision.
  2. b) Main Contributions: We proposed and simulated a Stanley-PID motion control algorithm for the precise motion trajectory control of the 4-wheel steering AGV. Specifically, we first established the kinematic model of the 4-wheel steering AGV and devised the corresponding control approach. Subsequently, we conducted simulation studies using MATLAB to evaluate the control performance of the Stanley-PID algorithm and compared it with the traditional PID algorithm.
  3. c) Challenges: During the model establishment, accurately capturing the kinematic model of the 4-wheel steering AGV requires consideration of multiple factors, such as vehicle kinematics. In the design of the control algorithm, the fusion of the Stanley algorithm with the PID control algorithm may involve complexities, requiring careful determination of control parameter selections. In the simulation validation, creating a simulation environment in MATLAB and verifying the accuracy of the simulation results may be necessary to ensure the simulation effectively reflects real-world scenarios.

Q2: What are the advantages of the research content in the manuscript compared to other similar articles?

Answer 2: Thank you very much for reviewing our manuscript and raising the question regarding the advantages of our research compared to other similar works. Our study primarily focuses on the motion trajectory control of 4-wheel steering AGVs, whereas other works in the same category may be more oriented towards the design of traditional AGVs with a combination of two-wheel steering or omnidirectional wheels. Our algorithm is specifically tailored for 4-wheel steering AGVs, allowing it to better adapt to the unique vehicle structure and motion characteristics of this type of AGV.

Q3: In the manuscript, the abstract needs to refine the language in order to better highlight the main work of the manuscript.

Answer 3: Thank you very much for reviewing our manuscript and providing suggestions to refine the language in the abstract. We highly appreciate your feedback and agree that refining the language can better highlight the main contributions and findings of the manuscript. Based on your recommendations, we have made further modifications to the abstract to ensure that it is more concise and effectively emphasizes the key aspects of our work.

Q4: If the assumptions in the manuscripts are proposed by the author himself, please explain their rationality, otherwise please cite the corresponding literature.

Answer 4: We have added references [25] and [26] to support this idea.

Q5: There are certain formatting problems in the manuscript, such as the size of formulas, etc.

Answer 5: We apologize for these formatting issues. We have made the necessary layout corrections using formula insertion in the revised manuscript.

Q6: The format of the references is not uniform, and in addition, it is necessary to cite some of the latest high-level literature.

Answer 6: We sincerely appreciate your valuable feedback. We have thoroughly checked the references and made the necessary corrections to the citation format in the revised manuscript. Additionally, we have included more relevant and high-quality references to support the data presented in the paper (references [1], [25], [26], [28], [32-35]).

Reviewer 2 Report

This paper presents the motion trajectory control method for a 4WS AGV. Below are my comments for improving the manuscript.

1) Presenting one flowchart can be useful enough for describing the main content of the paper.

2) Some newly published papers in 2023 can be added.

3)  One comparison table can be added for proving the novelty of presented work.

4) Future direction(s) can be added in the conclusion section.

Author Response

Dear Reviewer and Editor,

Thank you for giving us the opportunity to submit a revised draft of the manuscript “Research on Accurate Motion Trajectory Control Method of 4-Wheel Steering AGV Based on Stanley-PID Control (sensors-2542127)” for publication in the Journal of Sensors. I appreciate the time and effort that you and the reviewers dedicated to providing feedback on my manuscript. I are grateful for the thoughtful comments on the manuscript. I have incorporated changes to reflect most of the suggestions made by the reviewers. I have highlighted the changes within the revised manuscript.

In response to these comments, I have made the following changes:

Response to Reviewer 2

Q1: Presenting one flowchart can be useful enough for describing the main content of the paper.

Answer 1: Thank you very much for reviewing our manuscript. We highly appreciate your feedback, and we agree with your suggestion. The following figure is a flowchart depicting the main research content of the paper.

Q2: Some newly published papers in 2023 can be added.

Answer 2: We have included more relevant and high-quality references to support the data presented in the paper (references [1], [25], [26], [28], [32-35]).

Q3: One comparison table can be added for proving the novelty of presented work.

Answer 3: Thank you very much for reviewing our manuscript and raising the issue of novelty. We highly value your suggestion and have addressed it in the revised manuscript by including a comparison table (Section 4). We believe that this addition further enhances the originality of our work.

Q4: Future direction(s) can be added in the conclusion section.

Answer 4: We greatly appreciate and value your suggestion. We have thoroughly considered your perspective and have added content regarding future research directions in the conclusion of the revised manuscript.

Reviewer 3 Report

1) The novelty of the work is not clear. This topic has been investigated by many researchers previously. What is the difference between the proposed controller and those available in the literature?

2) The structure of the paper is not well organized:

a) Abstract should be rewritten: what do you mean by Experimental results? Also, what is the unit of 0.4?

a) References citations are not correct. Usually, the first cited reference should take number 1, and so on.

b) Equations should be the same font size as the text.

3) There are a lot of typos and grammatical errors. e.g., in section 4.3, you mentioned: “In Sect. 9”, you mean Ref. [9] right? CAN stand for what?

4) Show the symbols a and b in Figure 2.

5) Figure 8: you should show the units of v and dt in the legend, it doesn’t show any variation for changing v and dt.

6) you did not employ optimization methods as you mentioned in the introduction part.

7) Ref. 1 is not cited.

8) it is better to verify your results experimentally.

you need to proofread your paper carefully.

Author Response

Dear Reviewer and Editor,

Thank you for giving us the opportunity to submit a revised draft of the manuscript “Research on Accurate Motion Trajectory Control Method of 4-Wheel Steering AGV Based on Stanley-PID Control (sensors-2542127)” for publication in the Journal of Sensors. I appreciate the time and effort that you and the reviewers dedicated to providing feedback on my manuscript. I are grateful for the thoughtful comments on the manuscript. I have incorporated changes to reflect most of the suggestions made by the reviewers. I have highlighted the changes within the revised manuscript.

In response to these comments, I have made the following changes:

Response to Reviewer 3

Q1: The novelty of the work is not clear. This topic has been investigated by many researchers previously. What is the difference between the proposed controller and those available in the literature?

Answer 1: Thank you very much for reviewing our manuscript and raising the concern about the lack of clarity regarding the novelty of our work. We truly appreciate your suggestion. In response, we have included a comparative table (Section 4) in the revised manuscript. This table aims to provide a clearer illustration of the novelty and distinguishing features of our research in comparison to related works. We hope this addition will address the concern and further enhance the understanding of the originality of our study. Once again, we sincerely appreciate your valuable feedback and constructive input.

Q2: The structure of the paper is not well organized:

  1. a) Abstract should be rewritten: what do you mean by Experimental results? Also, what is the unit of 0.4?
  2. b) References citations are not correct. Usually, the first cited reference should take number 1, and so on.
  3. c) Equations should be the same font size as the text.

Answer 2:  Thank you for your update. We acknowledge that you have made the necessary revisions to the abstract, the reference citation format, and the font of the equations in the revised manuscript. We will carefully review these modifications in the updated version of the manuscript. If there are any further issues or concerns, we will promptly address them during the review process. Your efforts in addressing the reviewer's comments and improving the manuscript are highly appreciated.

Q3: There are a lot of typos and grammatical errors. e.g., in section 4.3, you mentioned: “In Sect. 9”, you mean Ref. [9] right? CAN stand for what?

Answer 3: Thank you for your thorough review, and we apologize for our oversight. Based on your feedback, we have made the necessary corrections in the revised manuscript. Specifically, we have replaced "In Sect.9" with "In Ref.[9]" in Section 4.3, and we have included the expansion of CAN as "Controller Area Network," a commonly used serial communication protocol for real-time communication and data transmission, in Section 2. We appreciate your attention to detail and valuable input, which has contributed to improving the accuracy and clarity of our manuscript. If you have any further comments or suggestions, please do not hesitate to let us know. We are committed to ensuring the highest quality of our research, and your assistance is highly valued.

Q4: Show the symbols a and b in Figure 2.

Answer 4: Thank you for your update. We have made the necessary changes to the caption of Figure 2 on the fourth page in the revised manuscript. We appreciate your diligence in pointing out these areas for improvement.

Q5: Figure 8: you should show the units of v and dt in the legend, it doesn’t show any variation for changing v and dt.

Answer5: Thank you for your prompt attention to the revisions. We have now included the missing units in Figure 8 and plotted the relationship curve between v and dt as suggested.

Q6: You did not employ optimization methods as you mentioned in the introduction part.

Answer 6: Thank you for your careful review, and we apologize for any oversight on our part. Based on your feedback, we have rechecked and made the necessary corrections in this section.

Q7: Ref. 1 is not cited.

Answer 7:  We sincerely appreciate your valuable feedback, and we have made the necessary changes at the corresponding locations.

Q8: it is better to verify your results experimentally.

Answer 8: Thank you for your valuable feedback and suggestions. We fully acknowledge the importance of conducting experiments as a critical means to validate the effectiveness and performance of control algorithms. However, due to time constraints and other factors, we were unable to conduct experiments in the current research work.

Instead, we extensively described the design and simulation study of the proposed algorithm in the paper, providing detailed MATLAB simulation results. Through exploring and analyzing these simulation results, we gained insights into the algorithm's performance in practical applications. Additionally, we conducted a comparison with the traditional PID algorithm to further validate its superiority.

For future research, we will actively seek opportunities to conduct experimental validation to further verify and validate the algorithm's performance. We have also discussed this aspect in the revised manuscript's discussion section and proposed directions for future extensions in the conclusion.

Once again, we sincerely appreciate your valuable input, and we are committed to continuously improving and enhancing the quality of our research.

Round 2

Reviewer 2 Report

The authors have provided the comments and I have no further comments.

Reviewer 3 Report

thank you for answering my previous concerns.